# Investigating the mechanism of impact of the Quality Premium initiative on antibiotic prescribing in primary care practices in England: a study protocol

Philip Emeka Anyanwu,[1] Sarah Tonkin-Crine,[2,3] Aleksandra Borek,[2] Ceire Costelloe[1]

[1]Global Digital Health Unit, Department of Primary Care and Public Health, Imperial College London, London, UK
[2]Nuffield Department of Primary Care Sciences, University of Oxford, Oxford, UK
[3]NIHR Health Protection Research Unit, Healthcare Associated Infections and Antimicrobial Resistance, University of Oxford, Oxford, UK

**Correspondence to**
Dr Philip Emeka Anyanwu;
anyanwuphilipemeka@gmail.com

## ABSTRACT

**Introduction** The persistent development and spread of resistance to antibiotics remain an important public health concern in the UK and globally. About 74% of antibiotics prescribed in England in 2016 was in primary care. The Quality Premium (QP) initiative that rewards Clinical Commissioning Groups (CCGs) financially based on the quality of specific health services commissioned is one of the National Health Service (NHS) England interventions to reduce antimicrobial resistance through reduced prescribing. Emerging evidence suggests a reduction in antibiotic prescribing in primary care practices in the UK following QP initiative. This study aims to investigate the mechanism of impact of this high-cost health-system level intervention on antibiotic prescribing in primary care practices in England.

**Methods and analysis** The study will constitute secondary analyses of antibiotic prescribing data for almost all primary care practices in England from the NHS England Antibiotic Quality Premium Monitoring Dashboard and OpenPrescribing covering the period 2013 to 2018. The primary outcome is the number of antibiotic items per Specific Therapeutic group Age-sex Related Prescribing Unit (STAR-PU) prescribed monthly in each practice or CCG. We will first conduct an interrupted time series using ordinary least square regression method to examine whether antibiotic prescribing rate in England has changed over time, and how such changes, if any, are associated with QP implementation. Single and sequential multiple-mediator models using a unified approach for the natural direct and indirect effects will be conducted to investigate the relationship between QP initiative, the potential mediators and antibiotic prescribing rate with adjustment for practice and CCG characteristics.

**Ethics and dissemination** This study will use secondary data that are anonymised and obtained from studies that have either undergone ethical review or generated data from routine collection systems. Multiple channels will be used in disseminating the findings from this study to academic and non-academic audiences.

### Strengths and limitations of this study

► This study will be the first to evaluate the mechanism of the impact of a financial incentive initiative involving Clinical Commissioning Groups (CCGs) to improve antibiotic prescribing in primary care practices in England.

► The investigation of multiple mediators in this study will help to identify the contributions of multiple strategies in translating the effects of Quality Premium (QP) while unpacking the extent of the effect of specific mediators.

► Due to the limited data on practice-level interventions or strategies that might potentially mediate the effect of the QP on antibiotic prescribing, we will not be able to extensively investigate the mechanism of QP impact at the practice level.

► Nevertheless, extensive investigations will be conducted at CCG level where the QP initiative is implemented and rewards paid out.

## INTRODUCTION

The persistent development and spread of resistance to antimicrobials, especially antibiotics, remain an important public health concern in the UK[1] and globally.[2,3] Antimicrobial resistance (AMR) is a major threat to the treatment and control of infectious diseases as drug-resistant infections are characterised by prolonged morbidity, increased risk of disabilities, death and cost of healthcare.[4] A 2014 review estimated that consistent increases in AMR would lead to about 10 million deaths per year by 2050.[5] Inappropriate prescribing and use of antibiotics in healthcare practices, especially primary care, are integral to the development and spread of resistance.[6,7]

About 74% of the antibiotics prescribed in England in 2016 was in primary care.[8] The high rate of antibiotic prescribing in primary care has been associated with increased AMR.[9] While some of the antibiotics prescribed in primary care settings are appropriate, a substantial proportion are cases where antibiotics are not clinically indicated, such as suspected respiratory tract conditions, which

BMJ

can be self-limiting.[9 10] Uncertainties about diagnosis, (perceived) patient expectations for antibiotics, occupational pressure (eg, consultation rate) and previous experiences are some of the identified drivers of overprescribing in primary care practices.[11–16]

Interventions such as antibiotic stewardship programmes, education and training initiatives targeted at prescribers and patients, financial incentives, among others have been implemented in England to reduce AMR through reduced prescribing. In particular, the Quality Premium (QP) is a National Health Service (NHS) England initiative established in 2013 to reward Clinical Commissioning Groups (CCGs) financially based on the quality of specific health services considered to be of national or local priority and commissioned over a specific period.[17] Improvement of antibiotic prescribing in primary care was one of the national priorities in the 2015/2016 guidance,[18] constituting 10% of the premium awarded from 2016/2017 to date.[19 20] Key aspects of the 'improved antibiotic prescribing' priority are reductions in the number of antibiotics prescribed in primary care facilities across England and in the proportion of broad-spectrum antibiotics prescribed in primary care (2015–2017).[19] Part of the requirements in the 2015/2016 QP guidance for demonstrating improved antibiotic prescribing by CCGs was a reduction in the number of antibiotics prescribed in primary care by 1% of the mean value in England in 2013/2014 (ie, 1.61 items per

Specific Therapeutic group Age-sex Related Prescribing Unit (STAR-PU)). This was further increased to 4% in the 2016/2017 guidance.

Emerging evidence suggests a reduction in antibiotic prescribing in primary care practices in the UK following the QP initiative.[21 22] Prescribing data from England show a reduction of about 2.7 million antibiotic items between 2014/2015 and 2016/2017 financial year.[23] Eighty-eight per cent of the CCGs in England achieved the target of reducing antibiotic prescribing in the first 2 years of QP.[21] Also, there has been a significant reduction in the proportion of broad-spectrum antibiotics prescription with 83% of the CCGs meeting their target in the first 2 years.[21] Such reductions in antibiotic prescribing would be expected to contribute to reductions in the development of resistance.[24] However, little is known about the mechanisms by which the QP initiative impacted on antibiotic prescribing in primary care practices.

The impact of interventions on specific outcomes is sometimes explained by a series of events. Potential mediators are important in assessing causal relationships like that between QP and antibiotic prescribing in primary care practices in England, where a potential mediator is a variable that hypothetically mediates the effect of QP on the outcome. The conceptual model shown in figure 1 demonstrates the hypothesised pathways for the impact of the QP initiative on antibiotic prescribing and subsequently AMR. The model is

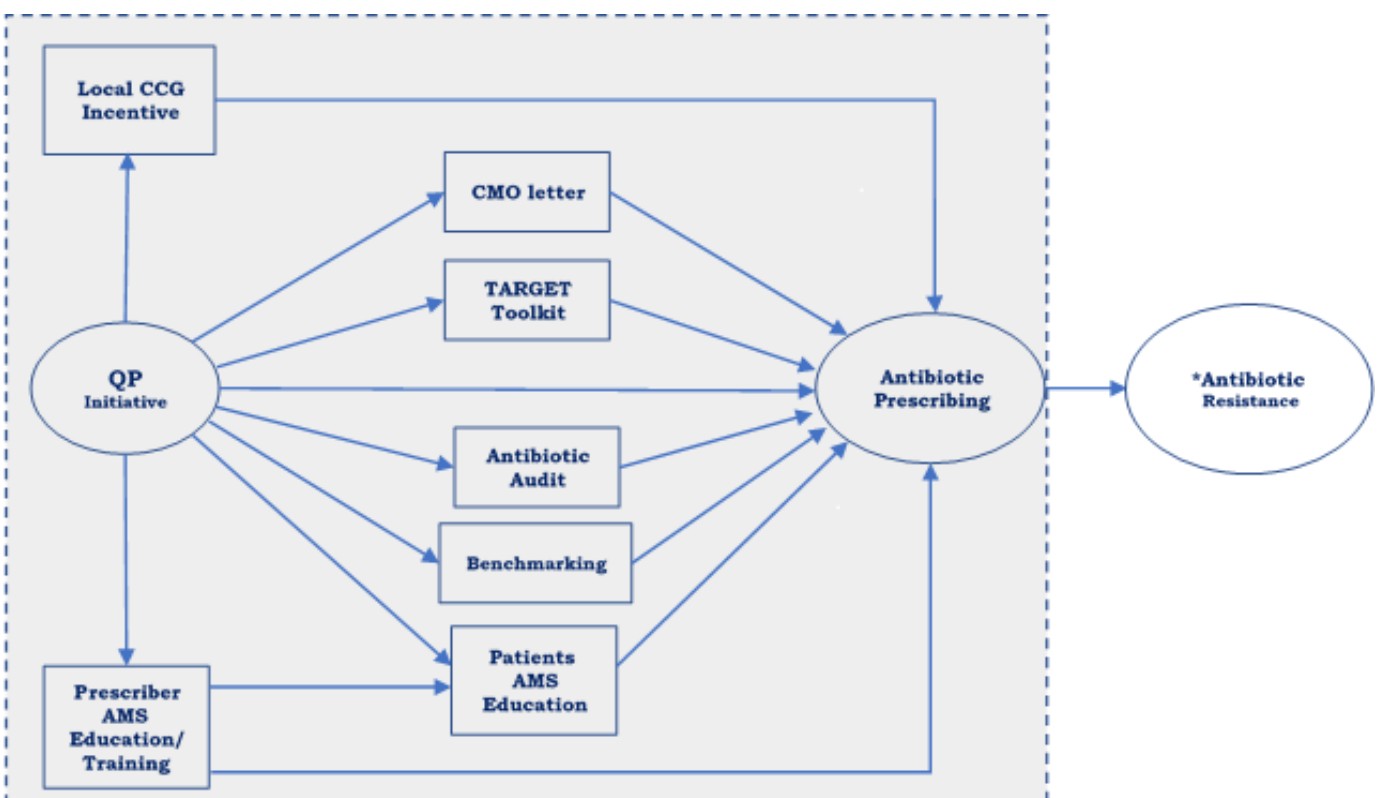

**Figure 1** Conceptual model. The direct effect is represented by the path between the QP initiative (predictor) and antibiotic prescribing rate (outcome). *Not directly observed within our data. AMS, antimicrobial stewardship; CCG, Clinical Commissioning Group; CMO, Chief Medical Officer; QP, Quality Premium; TARGET, Treat Antibiotics Responsibly, Guidance, Education, Tools.

developed based on conceptual and empirical evidence from existing literature and the results of qualitative and survey studies conducted as initial stages of the broader STEP-UP (Improving the uptake and SusTainability of Effective interventions to promote Prudent antibiotic Use in Primary care)[25] project that includes the current study. The conceptual model will be further validated through a stakeholder workshop with key antibiotic stewardship personnel, primary care prescribers and CCG representatives.

Our conceptual model suggests that in addition to its direct impact, the QP initiative acts by stimulating and enhancing the adoption of existing strategies to reduce and optimise antibiotic prescribing. In investigating the potential pathways connecting QP to reductions in antibiotic prescribing, we will be examining the hypothesis that factors like the Chief Medical Officer's (CMO) letter, Treat Antibiotics Responsibly, Guidance, Education, Tools (TARGET) toolkit, antibiotic auditing, benchmarking, local incentives at CCG level, prescribers and patients' antimicrobial stewardship (AMS) education/training can transmit part of the influence of the QP initiative to antibiotic prescribing.

First implemented in September 2014, the CMO letter, which provided social norm feedback to primary care practitioners in England whose antibiotic prescribing rate was in the top 20%, reduced antibiotic prescribing by 3.3% in 6 months in a randomised trial.[26] The criteria for selecting practices that received the letters in the subsequent years changed with the addition of measures like a change in antibiotic prescribing over time and whether practices have previously been sent CMO letter.

Another important mediator hypothesised in our conceptual model is AMS education/training of prescribers and patients. Educating and training prescribers on the importance of antibiotic stewardship and ways to promote prudence can help prescribers make better decisions on when an antibiotic is indicated and can also improve patients' knowledge on the appropriate use of antibiotics. AMS interventions targeted at patients can improve their knowledge of when antibiotics are not needed and increase confidence and skills on how to self-care, which can result in reduced consultations for self-limiting illness and thus reduced antibiotic prescriptions.

TARGET is a toolkit developed by the Public Health England (PHE), the Royal College of General Practitioners and other professional societies to promote prudent antibiotic use among prescribers and patients in primary care.[27] The intervention comprises of multiple resources (patient information leaflets on infection management and antibiotic use, self-assessment checklist for prescribers, antibiotic audit toolkits, interactive workshop presentations, national antibiotic management guidance, training resources and resources for clinical and waiting areas) to provide clinicians and patients with the motivation and skills to use antibiotics prudently.[27] A qualitative study evaluating prescribers' attitude and perception about the TARGET toolkit reported that general practitioners described it as useful and important in improving their prescribing behaviours and the expectations of their patients.[28] The use of resources like the TARGET antibiotics workshop has been shown to reduce antibiotic prescribing rate in a randomised controlled trial.[29]

We hypothesise that the implementation of the QP initiative informed the initiation or wider use of these strategies indicated in the conceptual model as mediators, which will subsequently influence antibiotic prescribing at primary care practices.

The implementation of the QP initiative in NHS England constitutes a natural experiment and offers an opportunity to investigate the preintervention and postintervention periods to understand the mechanism of impact of the QP intervention in reducing antibiotic prescribing rates in primary care practices.[30] Given the ethical and practical constraints in manipulating exposure in such an intervention, a natural experimental design offers a practical approach to understand the overall effect and mechanism of interventions like QP that offers financial incentives for clinical compliance. The publication of the 2015/2016 QP Guidance[18] constitutes the 'intervention', with periods before this as the 'control'.

## Study aim and objectives

Using routinely collected population-level data on antibiotic prescribing in England, this study aims to address the research question: What are the mechanisms and mediators of the impact of a high-cost health-system level intervention, the 'antibiotic prescribing quality premium'? We will investigate the difference in antibiotic prescribing rate pre-QP and post-QP initiative to establish its direct, indirect (through mediators) and total effects in reducing antibiotic prescribing in primary care practices in England.

## METHODS AND ANALYSIS
## Study design

Contemporary evaluations of the effectiveness of health policies go beyond estimating their total effect on outcomes. Mediation analysis decomposes the total effect of an intervention into separate causal pathways,[31] enabling an understanding of why and how policies work by estimating the direct and indirect effects of the exposure.[32] We will conduct mediation analyses investigating the potential mediators of the impact of QP on antibiotic prescribing in primary care in England, establishing the direct and indirect effects of the QP initiative.

## Data sources

The study will constitute secondary analyses of antibiotic prescribing data from NHS England covering the period 2013 to 2018. CCGs were established in England in April 2013 following the Health and Social Care Act 2012.[33] Data on antibiotic prescribing in primary care at CCG level

will be sourced from the NHS England Antibiotic Premium Monitoring Dashboard, which is produced by the NHS Business Services Authority (BSA).[34] Primary Care prescribing data are publicly available from the NHS BSA website. The dataset contains the number of antibiotic items (STAR-PU) prescribed in each CCG from the financial year 2015/2016 to 2018/2019, with data for October 2018 the latest at the time of this protocol. Data for the period 2013 to 2015 will be mapped to CCG level from the practice-level data.

Practice-level antibiotic prescribing data will be sourced from OpenPrescribing, an Evidence-Based Medicine DataLab project by the University of Oxford. OpenPrescribing publishes monthly antibiotics prescribing data from August 2013, with data for October 2018 the latest at the time of this protocol. Practice-level antibiotic prescribing data from OpenPrescribing is not STAR-PU weighted, so the extracted data will be STAR-PU weighted using figures from the 2013 Item-based age–sex weighting for oral antibacterials[14] and the number of registered patients in each age–gender category in a practice for each specific month.

Overall, the dataset offers coverage of at least 3 years postintervention given that antibiotic prescribing became a QP priority in March 2015. This will be important in investigating the immediate direct and indirect effects of the QP initiative while giving an insight into the sustainability of the identified effects (if any) in the long-term.

## Variables

The predictor will be a binary variable indicating the implementation of the QP intervention. The intervention will include all periods after March 2015 when the 2015/2016 QP guidance in England was published, while the control will be periods prior to this. The primary outcome of interest is the rates of antibiotic prescribing at CCG level in England, which will be a continuous variable indicating the number of items (per STAR-PU) prescribed per month.

To account for differences in practice and CCG characteristics that can contribute to variance in antibiotic prescribing, we will be adjusting for the number of general practitioners in each practice (from the NHS Workforce data),[35] the index of multiple deprivation (from the Department for Communities and Local Government),[36] prevalence of comorbidities (asthma, chronic obstructive pulmonary disease, diabetes, cancer, chronic kidney disease (from the NHS Quality and Outcomes Framework database)),[37] the prescribing rate of other non-antibiotic drugs (opioids and benzodiazepines) and seasonal influenza vaccination rate (from PHE).[38]

Mediator variables will be derived from the questionnaire data from a PHE survey,[39] with 187 of the 209 AMS leads representing CCGs in England and data from other organisations that have evaluated the interventions treated as potential mediators in this study. In the PHE survey, participants were required to complete questionnaire items, which included their adoption of national

and local strategies in their respective CCG to enable them to meet QP targets on antibiotic prescribing in primary care. The mediator variables will be binary or continuous variables, indicating the adoption of key interventions and intermediaries that are hypothesised to reflect in the integration of the QP guidance in improving antibiotic prescribing in primary care practices.

## Handling missing data

Over the period covered by this study (2013–2018), there have been changes in the number of practices and CCGs in England. Some practices have closed, new practices opened and some CCGs merged over the period; as such, we will have missing values for some observations. To maximise the use of existing observations, we will retain all observed values in the main analysis and impute the missing values using multiple approaches.[40] Missing values for the period before a new CCG was formed through merger of pre-existing CCGs will be imputed using the mean value of the CCGs that constitutes the merger; subsequently, these closed CCGs will be dropped from the dataset. Other missing values in this study will be handled using multiple imputation method on Mplus V.8.2. We will run a separate imputation model for the practice and CCG level datasets, the results will be averaged across 20 imputed datasets. Complete case analysis will be conducted as part of the sensitivity analyses for this study to examine the consistency of the results from the imputed set.

## Statistical analysis

Our first analysis will be an interrupted time series to investigate whether antibiotic prescribing rate in England has changed over time, and how such changes, if any, are associated with the QP implementation in March 2015. This will be conducted using the ordinary least square regression method[41] to assess whether the 2015/2016 QP establishment resulted in a shift in the level and trend in antibiotic prescribing in primary care in England compared with the period before the intervention. Using the practice-level dataset, a univariate time series with the mean antibiotic items (STAR-PU weighted) prescribed in all primary care practices in England for each month will be conducted using the *ITSA* function on Stata, with post-trend specification to show a postintervention trend. The Cumby-Huizinga general test for autocorrelation will be used for general specification test of serial correlation in the time series data.[42]

A new QP guideline was implemented each financial year with changes in the prescribing rate target and the proportion that improved prescribing constituted in the QP award for each year. Our mediation analyses will investigate the effects of the QP by comparing antibiotic prescribing rate in the financial year before its implementation to each subsequent year postimplementation; as such, our dataset for the mediation analyses will have the control group as the financial year before QP and the intervention group as a specific post-QP implementation

year. Three analyses will be conducted for each of the three financial years since QP establishment. This will enable us to compare the effects of the different target levels that have been set over the years and the proportions of the QP award attributed to improvement in antibiotic prescribing.

Using a unified model for the natural direct and indirect effects,[31] we will investigate the relationship between the QP initiative, the potential mediators and antibiotic prescribing rate with adjustment for the practice and CCG characteristics. This approach addresses the issues associated with the traditional approach to mediation analysis that obtains natural direct and indirect effect estimates through a non-trivial combination of parameter estimates from multiple models for the regression of the mediator and that of the outcome.[32 43] Also, the unified model is applicable to nonlinear regressions, different measurement types for outcome and mediator variables, and allows for interaction between the exposure and the mediator.[31]

We will first fit single-mediator models with each mediator separately modelled using the *medeff* function in Stata.[44] With the single-mediator models, we will be able to establish the individual influence of each potential-mediator variable. Variables that showed a mediating effect in the single-mediator models will be added to build a multiple-mediator model using a sequential mediation analysis method.[45]

A sequential multiple-mediator analysis is preferred to merely summing the effects of the single mediators because the sum may differ from the joint mediated effect, particularly as our potential mediators may influence one another.[45] Modelling all the significant mediators together provides a more accurate assessment of the mediation effects and causal relationship,[46 47] while assessing the indirect effect of a group of mediators in explaining how and why the intervention impacts on the outcome. The ordering of the mediators in the sequential mediation analysis will be based on evidence from the literature and the outcome of our stakeholders' workshop designed to identify possible pathways between the predictor, mediators and outcome. The workshop that will validate our conceptual model will also enable us to identify what mediators affect one another and inform interactions to include in our model. All analysis will be conducted in the Stata statistical package V.15.1.

### Sensitivity analyses
Sensitivity analyses using dummy implementation dates of 1 to 3 months before and after the actual month each of the QP guidance were published will be conducted to assess the difference between the time the guidance is published and the dissemination of information and development of local arrangements by the CCGs. These analyses will further help to investigate the anticipatory effect of the policy and whether lag in implementation attenuates the effect of the intervention.

Also, we will conduct separate analysis with the outcome variable (antibiotic prescribing rate) as a binary variable (indicating whether each CCG achieved the required rate of reduction in antibiotic prescribing as stated in the QP guidance for each year) to examine whether the classification of the outcome variable based on achievement of QP target influences the results.

Subgroup analyses will be conducted on clusters of primary care practices based on their antibiotic prescribing behaviour (high and low prescribers) and indices of deprivation to examine whether any effect of the QP initiative seen in the overall population is different in subgroups of practices. The top and bottom 20% antibiotic prescribers as of March 2015 will be categorised as high and low prescribers. To address the issue of regression to mean in subgroup analyses, we will build a separate model with categorisation into high and low prescribers based on the mean of the prescribing rate of practices in the last 3 months to March 2015. The use of mean of multiple measures will offer a better estimate of each practice's true mean before the 2015/2016 QP initiative.[48] The subgroup analysis based on the indices of deprivation data[36 49] at primary-care practice level will be important in establishing the equity impact of the QP initiative.

### LIMITATIONS
This study has some limitations. The survey that provided data on mediator variables included 187 of the 209 CCGs existing at the time of the study. However, this sample size is large enough for strong statistical power, and multiple imputation will be used to address issues on missingness. Furthermore, due to the limited data on practice-level interventions or strategies that might potentially mediate the effect of the QP on antibiotic prescribing, we will not be able to extensively investigate the mechanism of QP impact at the practice level. Nevertheless, extensive investigations will be conducted at CCG level where we have more data on potential mediators.

In 2013, the Primary Care Trusts in England, which were responsible for planning and commissioning healthcare services at the primary care level, were transformed to CCGs.[50] As such, our investigations are restricted to the CCG era. This is important as the QP initiative is implemented and rewards paid out at CCG level.

We recognise that the causal interpretation of any effect from our mediation analysis rests on assumptions such as sequential ignorability and exchangeability. Causal inference from this analysis will be limited given that our data are observational with the absence of random assignment of cases to treatment and mediator levels, as well as the likelihood of unmeasured confounders. The rate of consultation for conditions where antibiotics might be prescribed is one of the unmeasured confounders in our study. This has not been accounted for in our analyses as these data are not available nationally at the practice level.

## DISCUSSION

Although the QP intervention has been reported to have been effective in reducing antibiotic prescribing,[22] there remain important gaps in the evidence base for this intervention, especially in relation to its mechanism of impact. This study will be the first to evaluate the mechanism of the impact of a financial incentive initiative involving CCGs to improve antibiotic prescribing in primary care practices in England. If our study identifies some key mediators, like other interventions implemented in similar time or in response to the QP initiative, that explain the indirect effect of the QP intervention on antibiotic prescribing, this will provide important evidence on the effectiveness of the implementation of a package of interventions on antibiotic prescribing. The investigation of multiple mediators in this study will also help to highlight the contributions of multiple factors in translating the effects of QP while unpacking the extent of the effect of specific mediators.

Evidence on the mechanism of impact of strategies like QP will be important in improving its uptake and sustainability while maximising its potential in reducing antibiotic prescribing in primary care settings.

Finally, financially incentivised strategies for clinical compliance have been criticised for their ability to result in unintended consequences.[51] In the case of the QP initiative, unintended consequences like not prescribing antibiotics in cases where they are indicated are possible. However, some of the QP response strategies that we hypothesised as potential mediators (such as prescribers' AMS education/training) have the ability to mitigate this unintended consequence. By comparing the prescribing rate before and after the QP initiative and identifying the strategies that explain its effect, we will generate evidence that will be important in considerations of the future of this intervention and revisions that may help reduce potential unintended consequences.

### Patient and public involvement

This study will use secondary data mostly from routine collection system and will not directly involve patients or the public. The dissemination of the results will include communication channels and public engagement events that will involve primary care practitioners and CCG representatives.

### Ethics and dissemination

Prescribing data from NHS BSA and NHS Digital are generated from routinely collected prescribing data on items that have been dispensed in primary care practices in England. The survey that produced the data on mediator variables was registered with the PHE Research Support and Governance Office (RSGO) and approved by PHE Research Ethics and Governance Group (REGG) and Health Research Association (HRA).

Multiple channels will be used in disseminating the findings from this study to academic and non-academic audiences, which will include an engagement workshop with our stakeholder network, presentations in scientific conferences, publication in peer-reviewed journals and press conference coinciding with paper publications.

**Contributors** All authors contributed to the conception and study design. PEA and CC developed the analysis plan. The manuscript was drafted by PEA with further input from CC, ST-C and AB. All authors approved the submission of the manuscript.

**Funding** This study is funded by the Economics and Social Research Council (ES/P008232/1). ST-C received funding from the National Institute for Health Research Health Protection Research Unit (NIHR HPRU) in Healthcare Associated Infections and Antimicrobial Resistance at the University of Oxford in partnership with Public Health England (PHE) (HPRU-2012-10041). The funding body will not play any role in the design or conduct of the study. CC is supported via an NIHR Career Development Fellowship (NIHR-2016-09-015).

**Disclaimer** The views expressed are those of the author(s) and not necessarily those of the NHS, the NIHR, the Department of Health and Social Care or Public Health England.

**Competing interests** None declared.

**Patient consent for publication** Not required.

**Ethics approval** This study will use secondary data that are anonymised and obtained from studies that have either undergone ethical review or generated data from routine collection systems.

**Provenance and peer review** Not commissioned; externally peer reviewed.

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
