## [Reviewer comments · BMJ Open]

ARTICLE DETAILS

TITLE (PROVISIONAL)	Investigating the Mechanism of Impact of the Quality Premium Initiative on Antibiotic Prescribing in Primary Care Practices in England: A Study Protocol.
AUTHORS	Anyanwu, Philip; Tonkin-Crine, Sarah; Borek, Aleksandra; Costelloe, Ceire

VERSION 1 – REVIEW

REVIEWER	Lina Maria Ellegård Department of Economics, Lund University
REVIEW RETURNED	01-Apr-2019

GENERAL COMMENTS	* Overall, I feel that the study protocol gives the necessary details to grasp the planned analysis. However, I selected "No" on point 4 of the Reviewer Checklist (replicability) because at this stage of the project, it is not yet clear what the full model will look like. This is because the authors write that they will design causal ordering of the mediation model after having performed single mediator-analyses and literature searches (manuscript page 10, lines 285-287). I think that the current level of detail is acceptable for a study protocol. * However, on a related note, I don't really understand how the effect size from a single-mediator analysis is informative with respect to the causal ordering in the mediation model - the authors should be able to explain their reasoning already in the protocol. * When it comes to study limitations (Reviewer point 12), the authors only discuss data-related limitations. I think that two methodological issues should be more clearly acknowledged in the protocol (or at least in the final manuscript): i) The study is observational, thus even the main analysis of prescribing hinges on quite strong assumptions. ii) The mediation analysis relies on other strong assumptions, ie the sequential ignorability assumption (see Imai & Yamamoto, 2013, http://hdl.handle.net/1721.1/85869) * Regarding the different mediators, I would like to encourage the authors to consider explicitly how much overlap there is between the TARGET toolkit and prescriber AMS education. Could not the TARGET toolkit be viewed as part of prescriber AMS education?
---

REVIEWER	Eric Macy Kaiser Permanente San Diego United States Eric Macy is a partner in the Southern California Permanente Medical Group, is a member of the Ask An Expert Panel of the American Academy of Allergy, Asthma, and Immunology, has received research grants from ALK Abello, Inc. to study adverse drug reactions, and has served on clinical trial safety and monitoring committees for BioMarin, Ultragenyx, and Audentes.
REVIEW RETURNED	03-May-2019

GENERAL COMMENTS	We have noted in our healthcare program that when physicians are monitored for antibiotic use for a specific diagnosis, such as sinusitis, there is a reduction in the antibiotic use linked to the diagnosis of sinusitis, but also a lower rate of the diagnosis of sinusitis being made, and a higher rate of alternative diagnoses such as “bacterial infection” being made where antibiotics are then used. Please consider tracking the following global measures in your population as an important control to make sure there are not unintended consequences from your interventions.  1) Daily doses of antibiotics used per patient per year, by decade of life and by gender, over the entire study interval for all patients. 2) Rates of all the potential diagnoses of interest, that might be given antibiotics, and possible alternative diagnoses per patient per year, by decade of life and by gender, over the entire study interval, for all patients.
--

VERSION 1 – AUTHOR RESPONSE

Reviewer: 1

Comment: Overall, I feel that the study protocol gives the necessary details to grasp the planned analysis. However, I selected "No" on point 4 of the Reviewer Checklist (replicability) because at this stage of the project, it is not yet clear what the full model will look like. This is because the authors write that they will design causal ordering of the mediation model after having performed single mediator-analyses and literature searches (manuscript page 10, lines 285-287). I think that the current level of detail is acceptable for a study protocol.

Response: Thank you for highlighting this. As you mentioned, at the designing stage of the project, it is unclear what the full model with multiple mediators will look like as this will be dependent on the outcome of initial analysis from single-mediator models.

Comment: However, on a related note, I don't really understand how the effect size from a single-mediator analysis is informative with respect to the causal ordering in the mediation model - the authors should be able to explain their reasoning already in the protocol.

Response: We have updated the manuscript to clarify that the ordering of the mediators will be based on evidence from the literature and the outcome of our stakeholders' workshop designed to identify possible causal pathways between the predictor, mediators and outcome (see page 11 lines 290-295 in the revised version of manuscript). The workshop which will validate of our conceptual model, will also enable us to identify what mediators affect one another and inform interactions between variables in our model.

Comment: When it comes to study limitations (Reviewer point 12), the authors only discuss data-related limitations. I think that two methodological issues should be more clearly acknowledged in the protocol (or at least in the final manuscript): i) The study is observational, thus even the main analysis of prescribing hinges on quite strong assumptions. ii) The mediation analysis relies on other strong assumptions, ie the sequential ignorability assumption (see Imai & Yamamoto, 2013, <http://hdl.handle.net/1721.1/85869>)

Response: Thanks for highlighting this, the manuscript has been revised to address this (see page 13 lines 331-337). We recognise that the causal interpretation of any effect from our mediation analysis rests on assumptions such as sequential ignorability and exchangeability. Causal inference from this analysis will be limited given that our data is observational with the absence of random assignment of cases to treatment and mediator levels as well as the likelihood of unmeasured confounders. This has been added as a limitation in the manuscript.

Comment: Regarding the different mediators, I would like to encourage the authors to consider explicitly how much overlap there is between the TARGET toolkit and prescriber AMS education. Could not the TARGET toolkit be viewed as part of prescriber AMS education?

Response: As you highlighted, there is possible overlap and interaction between mediators. These overlaps will be verified in the stakeholders' workshop and explored in our models. Following the stakeholders' workshop, we anticipate to update our conceptual model to account for any identified overlaps between the potential mediators

Reviewer: 2

Comment: We have noted in our healthcare program that when physicians are monitored for antibiotic use for a specific diagnosis, such as sinusitis, there is a reduction in the antibiotic use linked to the diagnosis of sinusitis, but also a lower rate of the diagnosis of sinusitis being made, and a higher rate of alternative diagnoses such as “bacterial infection” being made where antibiotics are then used. Please consider tracking the following global measures in your population as an important control to make sure there are not unintended consequences from your interventions. 1) Daily doses of antibiotics used per patient per year, by decade of life and by gender, over the entire study interval for all patients. 2) Rates of all the potential diagnoses of interest, that might be given antibiotics, and possible alternative diagnoses per patient per year, by decade of life and by gender, over the entire study interval, for all patients.

Response: The reviewer’s recommendation on unintended consequences resulting from measures to reduce antibiotic prescribing is important and have been considered in previous studies. With regards to the Quality Premium Initiative, which is the main focus of our proposed study, recent studies examining the possibility of unintended consequence (such as changes in primary care consultation and hospital admission rates for diagnoses related to complications of both respiratory tract infection and urinary tract infection, uncomplicated infections including complicated intra-abdominal infection, complicated skin, as well as sepsis) have reported no significant association between the intervention and unintended clinical consequences. See Balinskaite et al., 2018 <https://academic.oup.com/cid/advance-article/doi/10.1093/cid/ciy904/5136397>

We are unable to control for the rate of potential diagnoses where antibiotics might be prescribed due to the unavailability of national data on consultation rate in primary care practice. This comes under the unmeasured confounders that we have now identified in our limitation (see page 13 lines 331-337)

VERSION 2 – REVIEW

REVIEWER	Lina Maria Ellegård (PhD) Department of economics Lund University Sweden
REVIEW RETURNED	20-Jul-2019

GENERAL COMMENTS	I’m satisfied with the revisions made by the authors.
---